# Diversity and Distribution of Hydrocarbon-Degrading Genes in the Cold Seeps from the Mediterranean and Caspian Seas

**DOI:** 10.3390/microorganisms13020222

**Published:** 2025-01-21

**Authors:** Yogita Warkhade, Laura G. Schaerer, Isaac Bigcraft, Terry C. Hazen, Stephen M. Techtmann

**Affiliations:** 1Department of Biological Sciences, Michigan Technological University, Houghton, MI 49931, USA; ymwarkha@mtu.edu (Y.W.); lgschaer@mtu.edu (L.G.S.); isbigcra@mtu.edu (I.B.); 2Department of Civil and Environmental Engineering, University of Tennessee, Knoxville, TN 37996, USA; tchazen@utk.edu

**Keywords:** 16S rRNA analysis, aerobic degradation, anaerobic degradation, bioremediation, Caspian Sea, cold seeps, hydrocarbon-degrading genes, Mediterranean Sea, metagenomic sequencing, microbial diversity

## Abstract

Marine cold seeps are unique ecological niches characterized by the emergence of hydrocarbons, including methane, which fosters diverse microbial communities. This study investigates the diversity and distribution of hydrocarbon-degrading genes and organisms in sediments from the Caspian and Mediterranean Seas, utilizing 16S rRNA and metagenomic sequencing to elucidate microbial community structure and functional potential. Our findings reveal distinct differences in hydrocarbon degrading gene profiles between the two seas, with pathways for aerobic and anaerobic hydrocarbon degradation co-existing in sediments from both basins. Aerobic pathways predominate in the surface sediments of the Mediterranean Sea, while anaerobic pathways are favored in the surface sediments of the anoxic Caspian Sea. Additionally, sediment depths significantly influence microbial diversity, with variations in gene abundance and community composition observed at different depths. Aerobic hydrocarbon-degrading genes decrease in diversity with depth in the Mediterranean Sea, whereas the diversity of aerobic hydrocarbon-degrading genes increases with depth in the Caspian Sea. These results enhance our understanding of microbial ecology in cold seep environments and have implications for bioremediation practices targeting hydrocarbon pollutants in marine ecosystems.

## 1. Introduction

Marine cold seeps are unique ecological niches on the ocean floor where hydrocarbons, such as methane and other organic substances, emerge due to geological processes [1,2,3]. These seeps are distributed globally, from shallow coastal regions to deep ocean trenches, providing environments where numerous microorganisms have adapted to thrive despite extreme conditions, including reduced temperatures, elevated pressure, high methane, hydrogen sulfide, and low oxygen and light levels [4,5,6]. The ecological and biogeochemical significance of cold seeps has driven research into the microbial communities that inhabit these environments. These microbial communities also have the potential to serve as reservoirs of important microbial metabolisms for potential applications in bioremediation and biotechnology [7,8].

Cold seeps are home to diverse chemosynthetic microorganisms that derive energy from the oxidation of inorganic molecules rather than from sunlight [1,2,3]. These environments also support various metabolic processes, including methane oxidation and sulfate reduction, carried out by key microbial groups such as methanotrophic bacteria, sulfate-reducing bacteria (SRB), and anaerobic methane-oxidizing archaea (ANME). These microbes have evolved specialized metabolic pathways to survive and thrive in the harsh conditions of cold seeps, utilizing both inorganic and organic compounds as energy sources [3,6,9,10,11]. For example, methanotrophic bacteria utilize methane as their primary energy source, preventing the potent greenhouse gas from escaping into the atmosphere. Sulfate-reducing bacteria have developed strategies to utilize sulfate as an electron acceptor enabling them to function in oxygen-scarce environments like cold seep sediments using the organic-rich fluids that are emanating from the seeps as electron donors. Similarly, anaerobic methane-oxidizing archaea can metabolize methane in the absence of oxygen, further contributing to the regulation of methane flux in marine sediments [1,10,12]. These microorganisms establish intricate ecosystems where various species fill distinct niches. Beyond their ecological importance, marine cold seeps intrigue scientists with the prospect of uncovering new species and unique biochemical pathways. Microorganisms from these environments often generate enzymes and substances with potential industrial or medicinal uses, spurring continued research into their possible applications [3,8,13,14,15].

Cold seeps also serve as natural laboratories for studying hydrocarbon degradation processes under aerobic and anaerobic conditions. Recent oil spills have introduced significant hydrocarbons into marine environments, prompting extensive research on the biodegradation capabilities of marine microbes [16,17]. Since most recent large marine oil spills have happened in oxic settings, much of this work has centered on understanding how these microbes process hydrocarbons in the presence of oxygen using oxygen as a terminal electron acceptor. However, in cold seeps, there can be sharp gradients of oxygen availability making cold seep sediments an interesting location to study the coexistence of both aerobic and anaerobic hydrocarbon degradation.

Aerobic hydrocarbon metabolism entails converting hydrocarbons like methane (CH_4_), ethane (C_2_H_6_), and propane (C_3_H_8_), as well as aliphatic and aromatic hydrocarbons into carbon dioxide and water. Various aerobic organisms utilize mono- and dioxygenases as part of the metabolic pathways for hydrocarbon degradation [1,6,18,19]. One well-studied enzyme in this context is alkane-1 monooxygenase, which catalyzes the oxidation of alkanes to primary alcohols, eventually producing fatty acids that are metabolized via beta-oxidation. This process relies on oxygen for the initial activation of hydrocarbons. There are also diverse dioxygenases involved in the biodegradation of aromatic hydrocarbons. In contrast, anaerobic hydrocarbon biodegradation involves different mechanisms that do not rely on oxygen. Anaerobic organisms use alternative compounds, such as fumarate, in the alkyl succinate synthase pathway to activate alkanes, allowing for their breakdown without the need for oxygen [20,21,22,23]. These processes are crucial for understanding the fate of hydrocarbons in anoxic marine sediments and the potential for natural attenuation of oil spills in anaerobic settings. The Deepwater Horizon oil spill in the Gulf of Mexico provided a significant case study for microbial hydrocarbon degradation [16]. Researchers identified specific bacteria capable of degrading crude oil components and studied their metabolic pathways [24,25,26,27]. These studies revealed that microbial communities could rapidly respond to hydrocarbon plumes, with specific taxa proliferating in response to the increased availability of hydrocarbons [17,18,28]. This adaptive response underscores the ecological significance of microbial hydrocarbon degradation and the potential for leveraging these processes in bioremediation efforts.

Numerous studies have investigated the diversity, functions, and ecological relationships of microorganisms in marine cold seeps [1,2,3,4,5,9]. Research on deep-sea sediments near cold seeps has revealed various microbial taxa, such as Desulfobacteria, Methanobacteria, Gammaproteobacteria, and other functional groups like SRB and ANME, many of which remain poorly understood. These studies have identified specific microbial groups adapted to the unique geochemical conditions of cold seeps, such as methanotrophic bacteria, SRB, and ANME [3,9,10,15]. These microorganisms play crucial roles in hydrocarbon degradation and nutrient cycling, with various genes and metabolic pathways associated with hydrocarbon processing identified in these microbes [29,30]. For instance, studies have highlighted the presence of genes encoding enzymes involved in the anaerobic oxidation of methane and sulfate reduction, which are essential for maintaining the biogeochemical balance in cold seep environments [9,20,31]. Additionally, the composition and distribution of microbial communities can vary significantly depending on their location. Specific bacterial groups are often found around particular sea vents, whereas more general patterns of microbial diversity and distribution are observed over broader regions [10,20,32,33,34]. Further research has shown that environmental factors such as depth, temperature, salinity, and the presence of hydrocarbons influence the microbial community structure in cold seeps and the biodegradation potential of marine microbial communities [3,10,32,35]. These studies underscore the complexity and ecological importance of cold-seep microbial communities, highlighting the need for continued exploration and characterization of these unique ecosystems.

The Caspian and Mediterranean Seas are among the largest landlocked water systems and have diverse cold seeps. Unlike the saline Mediterranean Sea, the Caspian Sea contains brackish water [36]. Both seas host diverse microbial communities crucial for nutrient cycling and maintaining ecosystem health [13,14,34]. SRB are extensively studied in these regions, showing variations in diversity and abundance based on environmental factors like salinity, temperature, and oxygen levels. Methanogenic archaea, another significant group, are influenced by similar environmental parameters and play a central role in the sediment’s carbon cycle. In the Mediterranean Sea, studies have shown that microbial communities associated with cold seeps exhibit significant diversity and functional specialization. These communities are adapted to the unique geochemical conditions of the Mediterranean, which include high salinity, variable oxygen levels, and elevated bottom water temperatures [33,37,38]. Methanotrophic bacteria and SRB dominate these environments, playing crucial roles in methane oxidation and sulfate reduction. Similarly, in the Caspian Sea, microbial communities are shaped by brackish water conditions, extensive hydrocarbon seepage, and in the southern basin, anoxic bottom water. These communities include diverse groups of bacteria and archaea involved in hydrocarbon degradation and nutrient cycling [34,39]. Moreover, some bacterial types are consistently found across various cold seep sites in the Eastern Mediterranean Sea, implying crucial roles in the health and operations of cold seep ecosystems. Factors like water depth, sediment type, and proximity to seep vents can influence the variety and composition of these bacterial communities.

Given the distinct environmental conditions in the Mediterranean and Caspian Seas, we expect that there will be different organisms encoding hydrocarbon-degrading genes in the Caspian and Mediterranean Seas. We hypothesize that the Eastern Mediterranean Sea, which is characterized by high salinity, low nutrient levels, and high oxygenation, will favor the proliferation of microorganisms that utilize aerobic metabolic pathways for hydrocarbon degradation. Aerobic hydrocarbon degraders, equipped with genes encoding enzymes such as alkane monooxygenases, cytochrome P450s, and dioxygenases, can rapidly metabolize hydrocarbons in the presence of oxygen [30,34,37]. In contrast, with its brackish and anoxic bottom waters, conditions in the southern basin of the Caspian Sea will select for microorganisms that thrive under anaerobic conditions. We expect these microorganisms will possess genes encoding enzymes like benzyl succinate synthase and fumarate reductase, which facilitate anoxic hydrocarbon degradation. Unique environmental selection in each sea results in distinct microbial community compositions and metabolic capabilities [10,17,32,35]. Understanding these differences is crucial for elucidating the biogeochemical roles of these microbial communities and their potential applications in the bioremediation of hydrocarbon pollutants in diverse marine environments.

Furthermore, we hypothesize that the diversity and distribution of hydrocarbon-degrading genes will not only vary with depth but will also differ between the Mediterranean and Caspian Seas, driven by their distinct environmental conditions. While it is well-established that sediment depth influences the diversity and distribution of hydrocarbon-degrading genes in cold seep environments [1,3,5,6] our study aims to build on this by exploring how these depth-related variations differ between the Mediterranean and Caspian Seas, which have unique environmental regimes. We hypothesize that the surface sediments in the Mediterranean, which are well-oxygenated due to direct contact with the oxygenated overlying water, will harbor a diverse array of microorganisms capable of aerobic hydrocarbon degradation [35]. These microorganisms are equipped with genes for oxygen-dependent enzymes, such as alkane monooxygenases and cytochrome P450s, which enable hydrocarbon metabolism [40]. As sediment depth increases, oxygen availability decreases, creating an anoxic environment that selects for anaerobic hydrocarbon degraders. In deeper sediments, microorganisms with genes encoding enzymes like benzyl succinate synthase and those involved in sulfate reduction or methanogenesis become more prevalent. This depth-dependent variation in gene distribution reflects the adaptation of microbial communities to the changing redox conditions with sediment depth. By conducting metagenomic analyses on sediment cores collected from different depths in the Mediterranean and Caspian Seas, this research aims to characterize the shifts in microbial community structure and functional potential. The primary objective is to investigate the diversity and distribution of hydrocarbon-degrading genes in microbial communities inhabiting cold seep sediments of these contrasting marine environments. Additionally, the study explores the role of depth in shaping microbial community structure and metabolic capabilities. A key goal is to provide insights into the biogeochemical processes governing hydrocarbon degradation in cold seep ecosystems and to inform strategies for mitigating hydrocarbon pollution through bioremediation.

## 2. Materials and Methods

### 2.1. Sample Collection and DNA Extraction

To test these hypotheses, a 16S rRNA and metagenomic analysis was conducted on sediment cores collected from the Mediterranean and Caspian Seas (Figure 1A). A core was collected from the North Alexandria Mud Volcano in the Eastern Mediterranean between 11 and 15 October 2012 (Figure 1B) [41]. This core was immediately frozen and transported back to the lab. Upon returning to the lab, this core was sectioned into 1 cm increments and portions frozen. A series of cores were collected from the southern basin of the Caspian Sea in 2013, as described in Mahmoudi et al., 2015 [34]. Cores were collected from two sites and sectioned at 4 cm intervals. The samples were then immediately stored at −80 °C until further analysis. Total genomic DNA was extracted, and DNA quality was assessed. DNA was extracted from Caspian Sea cores using a MolBio PowerSoil DNA extraction kit (Qiagen, Germantown, MD, USA). DNA was extracted from the Caspian Sea samples using the MP Biomedicals FastSpin soil DNA kit (Irvine, CA, USA).

### 2.2. 16S rRNA and Metagenomic Sequencing

The V3-V4 region of the 16S rRNA gene was amplified with the primer pair 341F/806R using the Zymo Quick-16S Plus NGS Library Prep Kit (Zymo Research, Irvine, CA, USA), which incorporates a 10 bp barcode for sample multiplexing. PCR conditions included an initial denaturation at 95 °C for 10 min, followed by 42 cycles of denaturation at 95 °C for 30 s, annealing at 55 °C for 30 s, and extension at 72 °C for 3 min, according to manufacturer instructions. Amplified products were verified via agarose gel electrophoresis. Amplicons were pooled and purified with magnetic beads included in the kit Libraries were quantified using a Qubit fluorometer (Thermo Fisher Scientific, Waltham, MA, USA). 16S rRNA sequencing was conducted on an Illumina NextSeq 2000 platform (San Diego, CA, USA) to generate 300 bp paired-end reads. Metagenomic libraries were prepared and sequenced on an Illumina NovaSeq S4 platform [34] using a 2 × 151 bp run.

### 2.3. 16S rRNA Data Processing and Analysis

Sequencing data were processed with the DADA2 package in R [42]. This included quality filtering and trimming of sequences to remove adapters and bases below a quality threshold, error rate modeling, and correction, denoising to infer amplicon sequence variants (ASVs), merging of overlapping paired-end reads, and removal of chimeric sequences. Taxonomic classification was performed using the SILVA database (version 138), which assigns taxonomy to the species level.

### 2.4. 16S rRNA Community Structure

Processed data were imported into the phyloseq package for further analysis [43]. The phyloseq object integrated the ASV table, taxonomic data, and sample metadata. Samples with fewer than 1000 reads were removed. ASVs with taxonomic assignment as mitochondrial or chloroplast DNA were excluded. Alpha (Shannon and Observed ASVs) and beta diversity metrics were calculated to assess microbial diversity and community composition differences across samples [44,45]. Ordination techniques, including principal coordinates analysis (PCoA) and permutational multivariate analysis of variance (PERMANOVA) using Bray–Curtis dissimilarity metrics, were employed to evaluate community differences statistically. Community membership was visualized at the phylum and genus levels using stacked bar plots. Abundances were normalized to relative abundances to facilitate comparisons between samples, providing clear insights into the microbial community of the sampled regions.

### 2.5. Metagenomic Data Processing and Analysis

The sequenced raw data was downloaded from the Joint Genome Institute for both Caspian and Mediterranean Sea samples. To ensure the quality of the metagenomic data, raw reads underwent a thorough quality control process using FastQC and Trimmomatic to remove low-quality bases and adapter sequences [46]. Kraken 2 was used to assess the taxonomic composition of the trimmed reads using the kraken2 Standard-8 database [47]. Quality-filtered reads were then assembled into contigs using MEGAHIT [48,49]. These contigs were binned into metagenome-assembled genomes (MAGs) using MetaBAT2, and the quality of the bins was assessed using CheckM version 1.1.3, focusing on completeness and contamination levels [50,51]. GTDB-Tk v2.4.0+ was then used for the taxonomic classification of the MAGs, improving the resolution and accuracy of the taxonomic assignments [52,53]. Contigs were annotated using the Contig Annotation Tool (CAT) to refine the accuracy of taxonomic assignments, allowing for the determination of taxonomic profiles and relative abundances [54].

Functional annotation of the MAGs was conducted using Prokka v1.14.5, which identifies genes and assigns putative functions based on homology searches against reference databases. Hydrocarbon-degrading genes were identified using the CANT-HYD Hidden Markov Models (HMM) [55]. These annotated genomes were further analyzed to identify the diversity and distribution of aerobic and anaerobic hydrocarbon-degrading genes.

In addition to analyzing Metagenome Assembled Genome bins, a non-binning analysis was conducted. Assembled metagenomic contigs were annotated through a similar pipeline. This non-binning analysis enabled the identification of genes in genomes that could not be effectively binned into MAGs initially, providing insights into microbial diversity beyond the initially binned genomes and facilitating the characterization of low-abundance microorganisms.

### 2.6. Metagenomic Statistical Analysis

The community structure in the Caspian Sea and Mediterranean Sea was assessed at the phylum and genus level using metagenomic data. Community structure was assessed using R, and ggplot2 was employed for data visualization [56]. Data from kraken2 taxonomic identification was combined into a table that listed the abundance of metagenomic reads based on genus-level data. These tables were imported into phyloseq for diversity analysis. Alpha diversity metrics, including Observed Species and the Shannon Index, were calculated using the phyloseq and vegan packages [43,44]. Beta diversity was analyzed through Bray–Curtis dissimilarity and visualized with Principal Coordinates Analysis (PCoA). Differences in community composition by sediment depth were statistically tested using PERMANOVA, and differences in alpha diversity between groups were evaluated with the Kruskal–Wallis test. All analyses were performed in R, with a significance threshold of *p* < 0.05.

To test our hypotheses, we compared the diversity and abundance of hydrocarbon-degrading gene categories between the two deep-sea basins. The annotation files from prokka were searched to identify the presence of aerobic hydrocarbon-degrading genes, which included aliphatic hydrocarbon degradation and aromatic hydrocarbon degradation (Appendix A). We also identified genes for anaerobic aliphatic hydrocarbon degradation and aromatic hydrocarbon degradation (Appendix A). Diversity was assessed by quantifying the number of genes in each metagenomic sample annotated as involved in hydrocarbon degradation. The number of genes associated with hydrocarbon degradation was identified in high- and medium-quality MAGs, with taxonomic classification data for these MAGs obtained through Kraken. High-quality MAGs are defined based on Bowers et al., 2017 (high-quality >90% completion and <5% contamination; Medium quality MAGs >50% completion and <10% contamination) [57]. This approach allowed us to identify the presence and diversity of hydrocarbon-degrading genes across different depths, providing insights into which genes are prevalent in the Caspian and Mediterranean Sea samples. This analysis enabled us to compare gene diversity between basins and depths; however, it should be noted that this is based on the identification of genes rather than detailed abundance mapping.

## 3. Results

### 3.1. Distinctions in Microbial Community Composition and Diversity Between the Mediterranean and Caspian Sediments

To test the hypothesis that there are distinctions in the microbial community composition between the two basins, we used 16S rRNA sequencing. Principal Coordinates Analysis (PCoA) based on Bray–Curtis dissimilarity of 16S rRNA ASV tables revealed distinct microbial community compositions between the Caspian and Mediterranean Seas (Figure 2). Caspian Sea samples (circles) clustered separately from Mediterranean Sea samples (triangles), with the majority of variation explained along Axis 1 (15.8%). This visual separation indicates that microbial communities differ substantially between the two locations. In the Caspian Sea, deeper samples (e.g., 28–35 cm) showed greater separation along Axis 1 and 2, suggesting increased diversity in deeper layers. In contrast, Mediterranean Sea samples exhibited more variation along Axis 2 (9.8%), but depth-related clustering was less pronounced than in the Caspian Sea (Figure 2A).

The visual differences observed in the PCoA plot were statistically confirmed by PERMANOVA analysis. A highly significant difference was found between microbial communities in the Caspian and Mediterranean Seas (*p* = 0.001, R^2^ = 12.4%), indicating that Location explains a substantial portion of the variation observed. While the PCoA suggested depth-related separation in the Caspian Sea, this observation was not statistically supported by PERMANOVA for depth (*p* = 0.573, R^2^ = 69.1%). In contrast, the Mediterranean Sea exhibited a significant depth effect on microbial community composition (*p* = 0.001, R^2^ = 24.5%), further supporting the trends seen in the PCoA plot (Figure 2A). The lack of significant depth-related variation in the Caspian Sea is most likely due to limited replication within depth strata, where only one sample was available per depth interval, in contrast to the multiple samples available for each depth grouping in the Mediterranean Sea.

To further explore the differences in community composition observed through multivariate analysis, we analyzed taxonomic differences in the 16S rRNA data. The microbial community structure varied with sediment depth at both the phylum and genus levels in the Caspian and Mediterranean Seas. Proteobacteria, including genera such as *Seep-SRB1* and *Woeseia*, were dominant across many sediment layers in the Caspian Sea, particularly at depths of 12–16 cm, 24–29 cm, and 28–35 cm, where microbial diversity was notably higher (Figure 2). This was followed by Desulfobacterota (e.g., *Desulfatiglans*), Bacteroidota, Chloroflexi (e.g., *Pelolinea*), and some members of Firmicutes, with *Candidatus Nitrosopumilus* also present across these layers (Figure 2B, Appendix A).

In the Mediterranean Sea, the microbial community was dominated by Proteobacteria, including *Coxiella* and Firmicutes, such as *Ammoniphilus*, with a lower presence of Carboxydothermus and Actinobacteriota also being present. There was a minimal presence of Bacteroidota and Desulfobacterota. The archaeal genus *Candidatus Nitrosopumilus* (from the phylum Thaumarchaeota) was also identified and is known for its role in ammonia oxidation [58]. Members of Subgroup 10 (from the phylum Chloroflexi) were also detected, though in lower abundance (Figure 2B, Appendix A). The results suggest that certain genera within Proteobacteria, such as Seep-SRB1 and Woeseia, play critical roles in Caspian and Mediterranean Seas biogeochemical processes. Seep-SRB1, known for its involvement in sulfate reduction, and Woeseia, which thrives in marine sediment environments, contribute to organic matter degradation and nutrient cycling in these ecosystems. Both seas exhibited significant microbial diversity across all sediment depths, with these key Proteobacteria genera contributing to the ecological functioning of these sedimentary environments.

### 3.2. Taxonomic Composition of Samples Based on Metagenomic Analysis

There are some known biases in 16S rRNA data related to the choice of primer pairs and amplification conditions. We used taxonomic analysis of metagenomic data to further explore the taxonomic differences between locations and with depth in these sediment cores. The data from metagenomic sequencing confirm many of the trends observed with 16S rRNA sequencing. Metagenomic analysis indicates that the microbial communities in the Caspian and Mediterranean Sea sediments were primarily dominated by Proteobacteria, Actinobacteria, and Firmicutes, with notable differences in the presence of other phyla. In the Caspian Sea, additional Bacterial phyla such as Bacteroidetes and Planctomycetes were common, whereas in the Mediterranean Sea, Bacteroidetes were present, but Planctomycetes were less abundant. The viral phylum Urioviricota was present in many of the samples from the Mediterranean and was highly abundant in deeper depths. Reads annotated as Chordata were present in many samples, which may be an indication of slight contamination. Proteobacteria remained the most abundant phylum across all depths in sediment cores from both seas, followed closely by Actinobacteria and Firmicutes (Figure 3).

At the genus level, notable differences were observed between the microbial communities of the two seas. *Streptomyces* and *Pseudomonas* were consistently present across all depths in the Caspian Sea, Followed by *Paenibacillus*, *Xanthomonas*, *Klebsiella*, and *Mycobacterium*, with *Burkholderia* appearing at some depths. Conversely, the Mediterranean Sea exhibited *Streptomyces*, *Valbivirus* as dominant genera, with *Klebsiella* being more abundant than the Caspian Sea. Notably, *Burkholderia* was absent in the Mediterranean Sea sediments. These genus-level distinctions provide a more detailed view of the niche specialization and adaptability at various depths that were indicated by the phylum-level analysis (Appendix A). Some differences were observed between the 16S rRNA sequencing and metagenomic data. These differences may be due to the use of different taxonomic databases, with Silva used for 16S rRNA and GTDB-Tk used for metagenomic analysis.

We also used the metagenomic data to assess diversity at the genus level (Figure 4). The observed genera count in the Caspian Sea showed a distinct pattern, with higher diversity in surface sediments and decreased diversity in deeper layers. This trend contrasts with the Mediterranean Sea, where the genera count displayed more variability across depths, reflecting a more complex depth-dependent microbial distribution. While the count of genera declined with depth in the Caspian Sea, the Shannon diversity index remained relatively stable, suggesting that although fewer genera were present at greater depths, the evenness of their distribution was maintained. In contrast, the Shannon index varied significantly in the Mediterranean Sea (*p* = 0.015), indicating more dynamic species richness and evenness shifts across different sediment layers.

Principal coordinate analysis (PCoA) revealed distinct clustering of microbial communities between the two seas, indicating clear separation based on geographic location (Figure 5). The clustering pattern highlighted that microbial communities in the Caspian and Mediterranean Seas are compositionally distinct at the genus level, further emphasizing the influence of unique environmental conditions on microbial structure in these regions. This finding confirms the 16S rRNA sequencing data. Furthermore, The PCoA plot also demonstrated depth-related differences in the microbial communities within each sea, with surface and deeper sediment samples forming separate clusters, reflecting the role of sediment depth in shaping microbial community structure (Figure 5). These findings again confirm using both 16S rRNA sequencing and metagenomic sequencing that there are key distinctions in the microbial communities in cold seeps in these two seas. Furthermore, these findings support the hypothesis that there are depth-related distinctions in the microbial community composition between these two sites.

### 3.3. Distribution of Aerobic and Anaerobic Hydrocarbon-Degrading Genes in Caspian Sea Sediments

The distribution of both aerobic and anaerobic hydrocarbon-degrading genes varied substantially across sediment depths in the Caspian Sea. Aerobic hydrocarbon-degrading genes were most abundant in the deeper sediment samples. Across all depths, the highest numbers of aerobic hydrocarbon-degrading genes were found in MAGs that were annotated as belonging to the Desulfobacterota (Figure 6). These genes primarily included alkane monooxygenases, such as, flavin-binding alkane monooxygenase, propane 2-monooxygenase, alkane oxidizing cytochrome P450, and long-chain alkane hydrolase, which are involved in the degradation of aliphatic hydrocarbons. Additionally, genes related to aromatic hydrocarbon degradation, such as dibenzothiophene monooxygenase, phenol hydroxylase, and benzene/toluene/naphthalene dioxygenase, were also prevalent, particularly in intermediate depths like 8–12 cm.

Additionally, the second most abundant aerobic hydrocarbon-degrading genes were found in MAGs annotated as Psedomonadota (Figure 6). Intermediate depths such as 8–12 cm depth in the Caspian also showed a high number of aerobic genes, while surface samples, particularly 0–4 cm and 4–8 cm in the Caspian, contained fewer aerobic genes with minimal taxonomic diversity. Asgardarchaeota with aerobic hydrocarbon-degrading genes were also shown to be present in deeper sediments. In contrast, the anaerobic hydrocarbon-degrading genes exhibited a different pattern, having the highest diversity in the mid-range depth sample, such as 8–12 cm in the Caspian core (Figure 6). This depth showed the highest number of anaerobic genes, primarily contributed by MAGs from the Desulfobacterota and Psedomonadota. Desulfobacterota and Psedomonadota are mostly abundant in all sediments, followed by Gemmatimonadota. Anaerobic degradation of aliphatic hydrocarbons was dominated by genes encoding alkyl succinate synthase and molybdopterin-family alkane C2 methylene hydroxylase, which were prevalent across multiple depths. Genes such as benzene carboxylase, naphthalene carboxylase, and benzyl succinate synthase were more abundant in anaerobic aromatic hydrocarbon degradation. Other depths, such as 4–8 cm and 30–35 cm, also contained relatively high counts of anaerobic genes, with significant contributions from diverse groups, including Acidobacteriota, Asgardarchaeota, and Chloroflexota. Surface sediments, such as 0–4 cm, displayed lower anaerobic gene counts, indicating a reduced capacity for anaerobic hydrocarbon degradation in these layers.

In the Caspian Sea sediments, aerobic aromatic hydrocarbon-degrading genes, such as dibenzothiophene monooxygenase, were found to be more prevalent across all sediment depths, followed by aerobic aliphatic-degrading genes like alkane oxidizing cytochrome P450, long-chain alkane hydrolase, flavin-binding alkane monooxygenase. For anaerobic hydrocarbon degradation, aliphatic-degrading genes like alkyl succinate synthase and molybdopterin-family alkane C2 methylene hydroxylase were consistently dominant at all depths, followed by anaerobic aromatic-degrading genes such as benzene carboxylase, molybdopterin-family ethylbenzene dehydrogenase subunit alpha, and naphthalene carboxylase. In Core 1 (0–4 cm, 4–8 cm) and Core 2 (0–4 cm, 4–8 cm, 8–12 cm), both aerobic and anaerobic hydrocarbon-degrading genes showed a consistent increase. However, between 12–16 cm, gene abundance decreased, followed by the rise again at the 32–35 cm depth level. In the Mediterranean Sea sediments, aerobic aromatic hydrocarbon-degrading genes, such as dibenzothiophene monooxygenase, are more abundant, along with genes involved in the degradation of long-chain alkanes.

Overall, these results highlight depth-related variations in both aerobic and anaerobic hydrocarbon degradation capacities in Caspian Sea sediments. Deeper and mid-range sediments appear to harbor more diverse and abundant microbial communities capable of hydrocarbon degradation, with aerobic processes more prominent in the deepest layers and anaerobic processes dominating the mid-range depths.

### 3.4. Distribution of Aerobic and Anaerobic Hydrocarbon-Degrading Genes in Mediterranean Sea Sediments

The distribution of aerobic hydrocarbon-degrading genes across the Mediterranean Sea sediment samples shows notable depth-related variation (Figure 7). The deepest layers (12–16 cm and 20–24 cm) exhibit the highest abundance of aerobic genes, indicating a higher potential for aerobic hydrocarbon degradation in these sediments. Pseudomonadota dominates in the 20–24 cm sample, playing a key role in aerobic degradation. Chloroflexota, Desulfobacterota, and Methylomirabiliota are significant in the 12–16 cm and 8–12 cm samples, contributing to aerobic processes at intermediate depths. Gemmatimonadota is moderately present in the 8–12 cm sample.

The distribution of anaerobic hydrocarbon-degrading genes across the Mediterranean Sea sediment samples shows substantial variation across depths. The layer (M1_8–12) exhibits the highest number of anaerobic genes, indicating a higher potential for anaerobic hydrocarbon degradation in these sediments. Desulfobacterota and Methylomirabilota dominate in the 8–12 cm, 12–16 cm, and 20–24 cm samples, playing key roles in anaerobic degradation processes. Gemmatimonadota and Chloroflexota also appear, contributing to anaerobic metabolism at intermediate depths. Planctomycetota is dominated in 0–4 cm depth, and Acidobacteriota becomes more dominant in the 16–20 cm sample. In the Mediterranean Sea sediments, aerobic aromatic hydrocarbon-degrading genes, such as dibenzothiophene monooxygenase, are more abundant, along with genes involved in the degradation of long-chain alkanes. For anaerobic degradation, aliphatic-degrading genes like molybdopterin-family alkane C2 methylene hydroxylase and alkylsuccinate synthase were detected, while aromatic degradation was primarily associated with benzene carboxylase. A trend was observed where aerobic and anaerobic hydrocarbon-degrading genes increase with depth in the Mediterranean Sea sediments.

Overall, both aerobic and anaerobic hydrocarbon-degrading genes were more abundant in deeper and mid-depth sediment layers, while surface layers exhibited fewer genes. Desulfobacteriota is a dominant contributor to both aerobic and anaerobic processes across all depths in both basins, highlighting its central role in hydrocarbon degradation in cold seep sediments. Desulfobacteriota genes include flavin-binding alkane monooxygenase, cytochrome P450, alkyl succinate synthase, and benzyl succinate synthase. Taxonomic diversity was observed, with different groups, contributing to hydrocarbon degradation at varying depths.

### 3.5. Analysis of Hydrocarbon Gene Abundance in the Binned and Non-Binned Metagenomic Data Across Caspian Sea Sediments

Analysis of both the binned and non-binned data reveals a predominance of aerobic hydrocarbon degradation in the Caspian Sea sediments, particularly in deeper layers (Figure 8). However, there are notable differences between the two approaches. The non-binned data should be more representative of the total diversity of hydrocarbon-degrading genes, but the binned data allows to link the diversity with taxonomic assignment for the MAG. In the non-binned analysis, aerobic genes showed extreme dominance in deeper layers like 20–24 cm, where gene counts exceeded 1428. This represents approximately 0.86% of the total genes in this layer (C2_20–24), indicating a high potential for aerobic hydrocarbon degradation. Anaerobic genes were more prevalent in mid-depth layers like 4–8 cm and 8–12 cm, though still secondary to aerobic processes. This analysis provides a detailed view of the raw gene counts, highlighting the stark differences in aerobic and anaerobic hydrocarbon degradation across sediment depths. The binned analysis highlights that while aerobic processes are dominant, anaerobic hydrocarbon degradation also plays a significant role, particularly in mid-depth to last layers where anoxic conditions are more favorable.

Overall, the combined results from both the nonbinned and binned analyses suggest that while aerobic hydrocarbon-degrading genes are more abundant across all depths in the Caspian Sea, and anaerobic degradation plays an important secondary role, especially in mid-depth sediments where oxygen is limited. The binned data provides a more moderated perspective on the relative contributions of these processes, while the nonbinned data emphasizes the large disparity between aerobic and anaerobic gene abundance. This finding was unexpected, given that the bottom waters of the Caspian Sea are largely anoxic, which typically favors anaerobic processes. This suggests that aerobic degradation may still occur or that facultative organisms can function in oxic and anoxic conditions.

### 3.6. Analysis of Hydrocarbon Gene Abundance in the Binned and Non-Binned Metagenomic Data Across Mediterranean Sea Sediments

The nonbinned and binned analyses of gene abundance in the Mediterranean Sea sediments reveal important insights into the distribution of aerobic and anaerobic hydrocarbon-degrading genes across different sediment depths. In the nonbinned analysis, anaerobic genes were more abundant than aerobic genes across most sediment depths (Figure 8). Mid-depth layers, such as 12–16 cm, 16–20 cm, and 20–24 cm, showed the highest abundance of anaerobic genes, exceeding 400 genes at some depths. In contrast, aerobic gene abundance was highest in the surface layer 0–4 cm, with over 500 genes. As depth increased, aerobic gene counts dropped significantly, indicating that anaerobic degradation processes dominate in the deeper sediment layers, while aerobic processes are more active in surface sediments where oxygen is more available. However, this observed difference in gene abundance by depth was not statistically significant (*p* = 0.905).

In the surface sediments (M1_0–4 cm), both aerobic and anaerobic gene counts were relatively low in the binned analysis, with around 15 genes each. This contrasts with the nonbinned data, where aerobic gene counts were significantly higher in the surface layer, suggesting that while both processes are present, their overall activity is more subdued when the data is binned.

Overall, the nonbinned analysis highlights the dominance of anaerobic hydrocarbon degradation in mid-depth and deeper sediments, while the binned analysis reveals a more co-occurrence of both aerobic and anaerobic gene abundance across all depths.

## 4. Discussion

The results of this study demonstrate distinct microbial community compositions and hydrocarbon degradation potential between the Caspian and Mediterranean Seas. Analysis of the taxonomic composition of the microbial community indicates significant differences between the two basins, with samples from each sea clustering separately on a PCoA plot. This finding indicates that location plays a major role in shaping microbial communities. These two seas have distinct geochemical contexts, with the Caspian being eutrophic, low salinity, and anoxic bottom waters [17,34,59]. Whereas the Mediterranean is oligotrophic (highly phosphate limited), with high salinity and elevated bottom water temperatures (14 °C) [41]. These distinctions in geochemistry are likely contributing to some of the observed differences in microbial community composition.

Depth-related variation in the microbial community composition was also observed throughout these sediment cores, particularly in the Mediterranean Sea. In the Mediterranean, microbial diversity, assessed using both Observed Species and the Shannon Index, increased in mid-depth sediments. The variability observed in the microbial communities of the Mediterranean Sea, indicated by the Shannon diversity index, suggests a greater microbial diversity which may reflect a more dynamic or variable environment compared to the Caspian Sea [60,61]. This trend aligns with Mahmoudi et al. [34], who also observed depth-related variations in microbial communities in the Caspian Sea, highlighting the role of depth in structuring microbial diversity. Similarly, other studies from the Mediterranean Sea [41,62] have shown significant depth-dependent differences in microbial community composition influenced by unique geochemical characteristics such as high salinity and oxygen availability. PCoA and PERMANOVA analyses confirmed significant differences in microbial community structure between the Caspian and Mediterranean Seas, highlighting the influence of distinct environmental conditions in shaping these communities. Depth-related stratification was also observed, driven by factors such as nutrient availability, oxygen levels, and pressure gradients, further supporting the role of environmental factors in microbial community structure.

The 16S rRNA gene analysis revealed a diverse microbial community in both basins, with variations linked to depth and environmental conditions. Key microbial taxa identified included *Seep-SRB1* and *Woeseia*, *Desulfatiglans*, and *Pelolinea* across different Caspian Sea depths, while *Coxiella*, *Ammoniphilus*, *Carboxydothermus*, Candidatus *Nitrosopumilus*, Members of Subgroup 10 were prominent in the Mediterranean Sea. A total of 15 high-quality MAGs were recovered from the Mediterranean Sea and 99 high-quality MAGs from the Caspian Sea, confirming the functional potential of these microbial taxa across different depths (S1-Spreadsheet). The presence of key microbial taxa at different depths aligns with previous findings from other cold seep environments, where depth plays a crucial role in determining microbial structure and function [6,13]. This depth-related variation is likely influenced by multiple factors, with oxygen availability being one of the key drivers. However, other factors, such as nutrient gradients and temperature changes, may also significantly shape microbial community structures across different depths.

The metagenomic analysis also revealed a diverse microbial community, including genera such as *Streptomyces*, *Pseudomonas*, *Paenibacillus*, *Xanthomonas*, *Klebsiella*, *Mycobacterium*, and Burkholderia. Taxa such as *Streptomyces*, *Pseudomonas*, and *Burkholderia* were identified across various depths, showing depth-related variation in community structure and function. This is consistent with findings from Mahmoudi et al. (2015), who reported similar taxa in the Caspian Sea, as well as Griffiths et al. (2023), who found that Gammaproteobacteria and other hydrocarbon-degrading taxa played a significant role in these environments [34,59]. These studies demonstrate that depth affects microbial diversity and drives the selection of specific metabolic functions necessary for survival and hydrocarbon degradation under varying oxygen conditions. The results indicate that mid-depth and deeper sediments harbor distinct microbial communities, particularly those associated with hydrocarbon degradation. The results indicate that mid-depth (approximately 8–25 cm) and deeper sediments (below 25–35 cm) harbor distinct microbial communities, particularly those associated with hydrocarbon degradation. These depth ranges are relative to the core length used in this study.

A clear difference between the Caspian and Mediterranean basins was observed in the distribution of aerobic and anaerobic hydrocarbon-degrading genes. Aerobic hydrocarbon-degrading genes were more diverse in the deeper sediments of the Mediterranean Sea and the Caspian Sea. In contrast, more anaerobic hydrocarbon-degrading genes were found in the Caspian Sea, especially in mid-depth sediments where conditions are expected to be anoxic, favoring anaerobic metabolism. This highlights the impact of environmental factors, such as oxygen availability, on the metabolic capabilities of microbial communities. The finding that aerobic hydrocarbon-degrading genes were so abundant in sediments of the Caspian is interesting due to the low oxygen content of the bottom waters above these sites in the Caspian [34]. The high diversity of aerobic hydrocarbon-degrading genes is indicative of genetic potential rather than activity. Thus, the elevated diversity of aerobic hydrocarbon-degrading genes in these samples could just indicate the presence of non-active bacteria in these sediments. Additionally, previous studies on the oil biodegradation potential of water above these sites indicated that relatives of aerobic hydrocarbon-degrading bacteria such as the Oceanospirillales and members of the *Alteromonas* as well as relatives of the *Thaumarchaeota* dominate the water samples in these Caspian locations despite the anoxic conditions [17]. Therefore, the presence of aerobic hydrocarbon-degrading genes, while interesting in light of the anoxic conditions of the bottom water, supports previous work on the deepwater microbial communities of the Caspian Sea.

In the present study, distinct MAGs encoding hydrocarbon-degrading genes were identified in both basins. Aerobic genes were predominantly associated with Desulfobacterota and Psedomonadota in the Caspian Sea, while anaerobic genes were linked to Desulfobacterota, Psedomonadota, and Gemmatimonadota. In the Mediterranean Sea, aerobic genes were linked to Desulfobacterota and Methylomirabiliota, whereas anaerobic genes were primarily associated with Desulfobacterota, Methylomirabilota, Gemmatimonadota, and Chloroflexota. These phyla are known for their adaptability to various oxygen levels and ability to degrade aliphatic and aromatic hydrocarbons. While Desulfobacterota are often sulfate-reducing bacteria, previous work has shown that they can tolerate some oxygen [63]. Furthermore, they are commonly considered anaerobic hydrocarbon-degrading bacteria in marine sediments [64]. Previous work has demonstrated that many of them encode and express anaerobic hydrocarbon-degrading genes (such as *assA*, *bssA*, and *badA*) [65].

The hydrocarbon-degrading pathways identified in this study further highlight the metabolic diversity of microbial communities. Aerobic pathways such as alkane oxidizing cytochrome P450 and long-chain alkane hydrolase were prevalent in the Mediterranean Sea, indicating an active capacity for aliphatic hydrocarbon degradation under oxic conditions [40,55]. Dibenzothiophene monooxygenase, responsible for aerobic aromatic degradation, was also present in both basins. In contrast, the Caspian Sea showed a higher abundance of anaerobic hydrocarbon-degrading genes, such as alkyl succinate synthase (AssA) and molybdopterin-family alkane C2 methylene hydroxylase (AhyA), reflecting the anoxic conditions of its mid-depth sediments. Benzene carboxylase and naphthalene carboxylase were also identified, highlighting the capacity for aromatic hydrocarbon degradation under anoxic conditions [55].

Furthermore, the unbinned and binned data provided additional insights into the functional potential of these microbial communities. The unbinned data showed distinct trends in the abundance of hydrocarbon-degrading genes across both basins. In the Caspian Sea, aerobic genes were generally more abundant in the deeper sediments, suggesting a higher potential for aerobic hydrocarbon degradation in these layers. In contrast, anaerobic genes showed significant abundance in mid-depth sediments (e.g., C1_4–8 and C2_8–12), reflecting adaptation to anoxic or low-oxygen conditions prevalent at these depths. In the Mediterranean Sea, anaerobic genes were more dominant across most depths, indicating that anaerobic hydrocarbon degradation processes were more prominent in this basin. Aerobic genes were present but generally less abundant, suggesting the potential for aerobic hydrocarbon degradation was more limited. In the Mediterranean, aerobic hydrocarbon-degrading genes decreased as depth increased, while anaerobic hydrocarbon-degrading genes remained at high levels. This overall trend of higher anaerobic genes in deeper depths highlights the adaptation of microbial communities to oxygen-limited conditions in deeper sediment depths.

The combination of unbinned and binned data provides a comprehensive understanding of the functional dynamics of these microbial communities. While the unbinned data revealed a broader range of hydrocarbon-degrading capabilities across different sediment depths, the binned data allowed for the specific identification of microbial taxa responsible for these functions. However, challenges with metagenome recovery suggest that there may be unexplored or undetected organisms in these environments, highlighting the complexity and potential for further discovery in cold-seep microbial communities. This dual approach strengthens our understanding of how microbial communities in cold seep environments contribute to hydrocarbon cycling and overall ecosystem function, emphasizing both the diversity of genes present and the specific metabolic pathways utilized by different taxa.

In this study, we found that the Mediterranean Sea had more aerobic hydrocarbon degradation in the deeper layers, while the Caspian Sea relied more on anaerobic processes, especially in mid-depth sediments. These differences are likely due to the unique geochemical features of cold seeps, such as varying oxygen levels, nutrient availability, and hydrocarbon presence. This pattern aligns with what is known about cold seep ecosystems—these areas are essential for breaking down hydrocarbons and contribute significantly to processes like methane cycling. Cold seeps provide a unique environment where microorganisms interact with different chemical conditions. In our study, vital microbial taxa, such as Pseudomonas, Streptomyces, and Desulfobacterales, were identified, all known for their ability to degrade hydrocarbons under harsh conditions [6,31]. Similar microbial activities have been observed in other cold seep areas, showing how these microbes help reduce hydrocarbon emissions naturally. The presence of these microorganisms in both the Caspian and Mediterranean Seas highlights their ability to adapt to the unique geochemical conditions in cold seep environments [13].

Overall, our findings add to the growing knowledge about cold seep ecosystems. These environments do not just support a wide range of microbes; they are also crucial for global carbon cycling and controlling hydrocarbons in the ocean. Understanding how these communities function helps us learn more about natural processes that help maintain balance in marine environments. Future research should continue to explore these unique ecosystems to understand their role in hydrocarbon cycling and the health of the broader marine environment.

### Applications in Bioremediation

The results of this study demonstrate that microbial communities in cold seep environments have significant potential for use in bioremediation of hydrocarbon pollution. The presence of both aerobic and anaerobic hydrocarbon-degrading genes in the Caspian and Mediterranean Seas highlights their adaptability to various environmental conditions, including oxygen availability and sediment depth. These microbial communities could be critical in breaking down hydrocarbons in marine environments, such as those impacted by oil spills. By understanding the distribution of hydrocarbon-degrading genes and their associated microbial taxa, we can explore ways to enhance natural degradation processes. This knowledge could be applied to develop more effective strategies for mitigating the environmental impacts of hydrocarbon pollution and supporting ecosystem recovery.

## 5. Conclusions

This study highlights the significant differences in microbial community composition and hydrocarbon degradation potential between the Caspian and Mediterranean Seas. Depth and environmental conditions, particularly oxygen availability, played crucial roles in determining the distribution of aerobic and anaerobic hydrocarbon-degrading genes. The Caspian Sea showed a greater reliance on anaerobic processes in mid-depth sediments, while the Mediterranean Sea exhibited more aerobic degradation in its deeper sediments.

The findings from the 16S rRNA gene analysis and hydrocarbon-degrading gene distribution offer valuable insights into how microbial communities in cold-seep environments adapt to varying environmental conditions. The use of both unbinned and binned data allowed for a comprehensive understanding of the functional capabilities of these communities. This understanding can be crucial for bioremediation efforts, as it helps identify which microbial communities are best suited for breaking down hydrocarbons under different environmental conditions, such as low oxygen availability. Further research is needed to explore the long-term ecological impacts of hydrocarbon degradation in these unique marine ecosystems and to investigate how changes in environmental factors, such as oxygen levels and nutrient availability, influence microbial community dynamics and their potential for oil biodegradation. Understanding these microbial processes better could lead to more effective natural bioremediation strategies for managing oil spills and reducing hydrocarbon pollution in marine environments.

Future research could build on this work by experimentally testing the metabolic pathways identified in this study. Studying the genetic and functional diversity of uncultured microorganisms could help discover new pathways with potential uses in cleaning up pollution and other biotechnological applications.

## Figures and Tables

**Figure 1 microorganisms-13-00222-f001:**
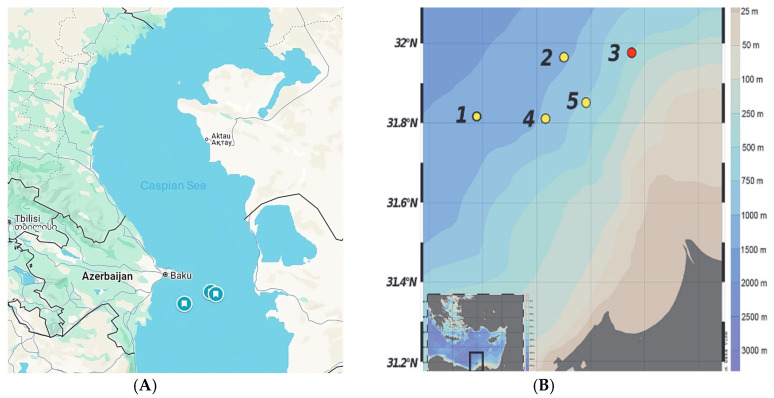
Map of Sampling locations. (**A**) Caspian Sea Sampling Zones. (**B**) Station 3 North Alexandria Mud Volcano, Mediterranean Sea other station numbers represent additional stations sampled in Techtmann et al 2015 [41].

**Figure 2 microorganisms-13-00222-f002:**
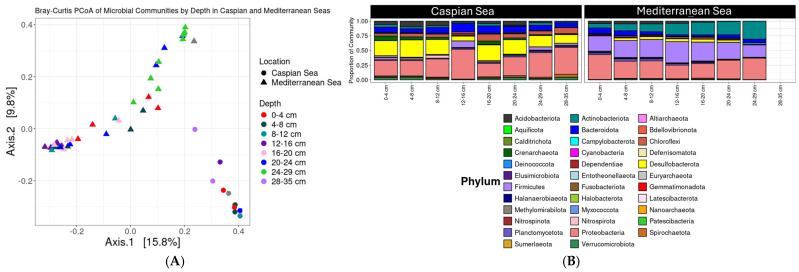
Microbial community diversity data based on 16S rRNA sequencing. (**A**) Principal coordinate analysis (PCoA) of microbial communities by depth in Caspian and Mediterranean Seas Based on Bray–Curtis dissimilarity. Samples from the Caspian are shown as circles, and the Mediterranean are shown as triangles. Colors represent depth groupings based on cm below the seafloor. (**B**) Relative abundance of microbial phyla in Caspian and Mediterranean Sea sediment samples.

**Figure 3 microorganisms-13-00222-f003:**
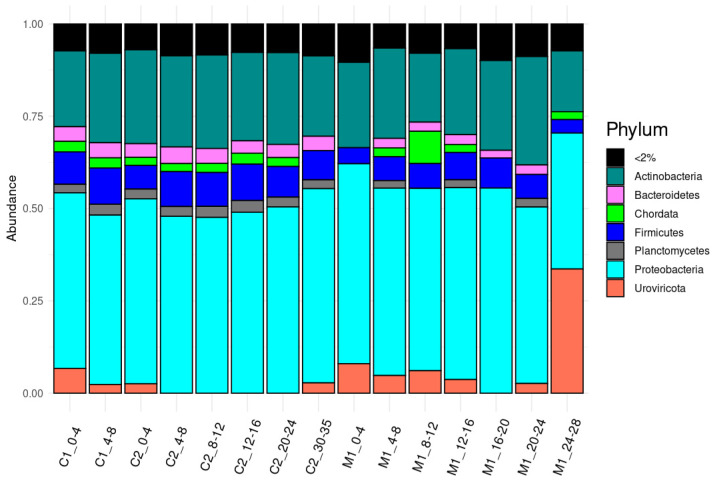
The taxonomic distribution Phylum in the Mediterranean Sea (M1) and Caspian Sea (C1—Core 1, C2—Core 2) sediment samples.

**Figure 4 microorganisms-13-00222-f004:**
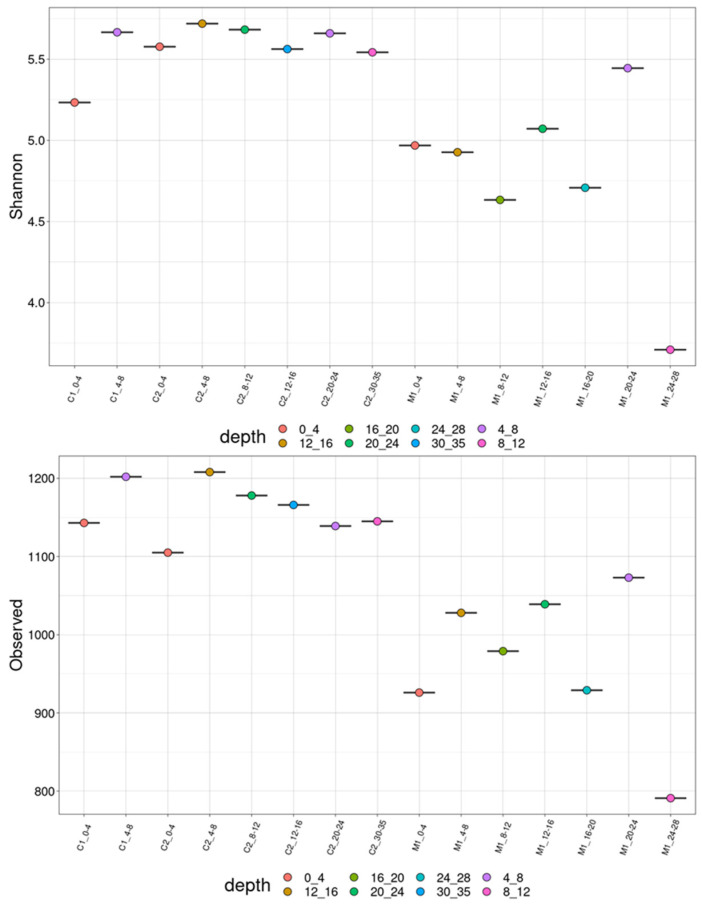
Shannon diversity and observed species richness at the genus level in Caspian (C1) and Mediterranean (M1) sea sediments.

**Figure 5 microorganisms-13-00222-f005:**
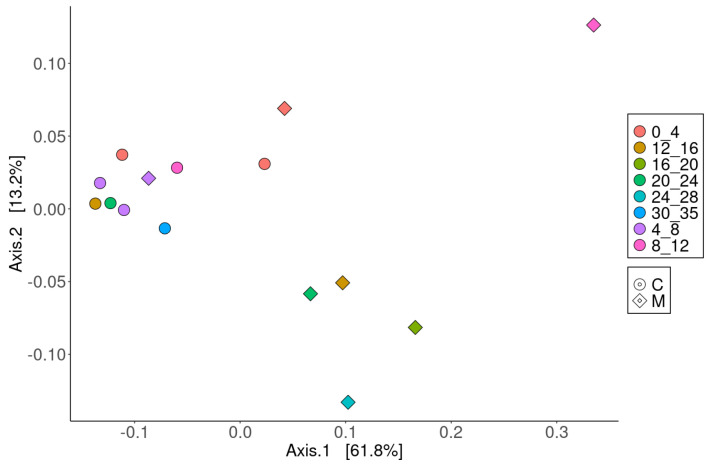
Principal coordinate analysis (PCoA) at the genus level for microbial communities in Caspian and Mediterranean Seas.

**Figure 6 microorganisms-13-00222-f006:**
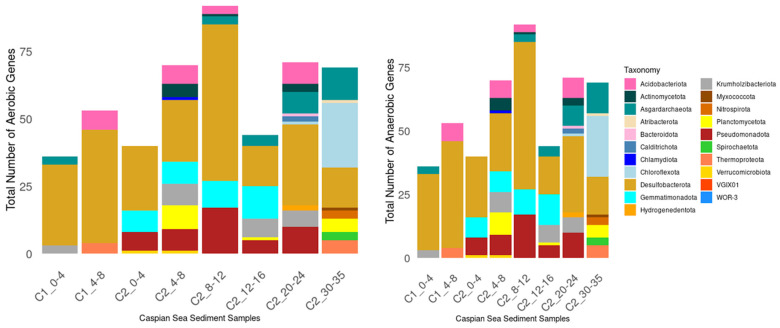
Distribution of aerobic (**Left**) and anaerobic (**Right**) hydrocarbon-degrading genes in Caspian Sea metagenomes.

**Figure 7 microorganisms-13-00222-f007:**
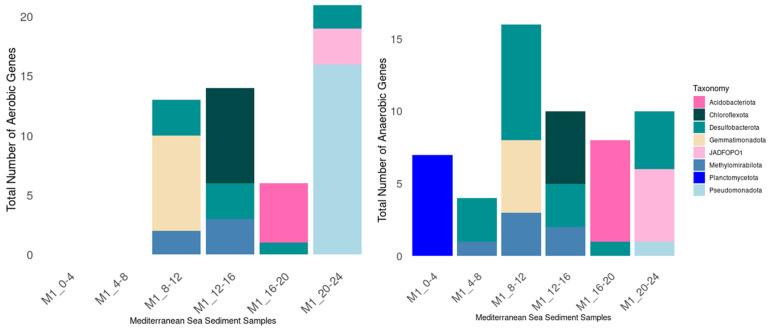
Distribution of aerobic (**Left**) and anaerobic (**Right**) hydrocarbon-degrading genes in Mediterranean Sea metagenomes.

**Figure 8 microorganisms-13-00222-f008:**
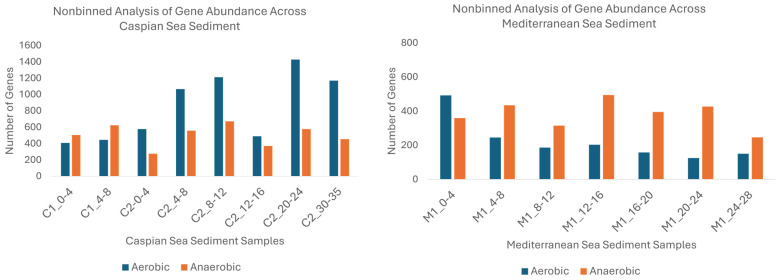
Non-binned analysis of gene abundance across Caspian Sea (**Left**) and Mediterranean Sea (**Right**) sediments.

## Data Availability

16S rRNA sequencing raw reads are deposited at the NCBI SRA associated with BioProject PRJNA1201138. Metagenomic data is depositied at the JGI IMG associated with the following IMG Genome IDs: 3300048500, 3300048081, 3300048537, 3300048080, 3300048501, 3300048058, 3300048070, 3300048060, 3300048840, 3300048498, 3300048076, 3300048495, 3300048079, 3300048055, 3300048077, 3300048075, 3300048499, 3300048496, 3300048990, 3300048057, 3300048059, 3300048056, 3300048497, 3300048494. All R scripts are available online on Github page at https://github.com/ymwarkha (accessed on 21 December 2024).

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
