# Peer review of "Diversity and Distribution of Hydrocarbon-Degrading Genes in the Cold Seeps from the Mediterranean and Caspian Seas"

_microorganisms, 2025, doi:10.3390/microorganisms13020222_

Round 1
Reviewer 1 Report
Comments and Suggestions for Authors
This manuscript aligns closely with my area of expertise. After reviewing it, I did not find any significant issues. However, there are a few minor typos, such as "16S," which can be easily corrected. One major concern, though, is data availability. While the authors indicate that sequencing data is available at the Joint Genome Institute’s Genome Portal (https://genome.jgi.doe.gov/portal/Divofcarcycling/Divofcarcycling.info.html), I was unable to access it. I recommend that the authors deposit the 16S and metagenomic raw reads into the NCBI database to ensure broader accessibility and compliance with standard data-sharing practices.
Author Response
Summary :
Thank you very much for taking the time to review this manuscript. Please find the detailed responses below and the corresponding revisions/corrections highlighted/in track changes in the re-submitted files.
Comment : This manuscript aligns closely with my area of expertise. After reviewing it, I did not find any significant issues. However, there are a few minor typos, such as "16S," which can be easily corrected. One major concern, though, is data availability. While the authors indicate that sequencing data is available at the Joint Genome Institute’s Genome Portal (https://genome.jgi.doe.gov/portal/Divofcarcycling/Divofcarcycling.info.html), I was unable to access it. I recommend that the authors deposit the 16S and metagenomic raw reads into the NCBI database to ensure broader accessibility and compliance with standard data-sharing practices.
Response 1 : Thank you for your comment. We reviewed the manuscript and found "16 S" written on lines 186, 198, and 205. We have corrected it to "16S" to ensure accuracy and consistency.
Response 2 : We agree and apologize for challenge with accessing the link. We have uploaded the 16S rRNA reads to the SRA at NCBI. We have now listed the JGI IMG Genome IDs for the metagenomes as the metagenomes were sequenced at JGI and the metagenome IDs include additional analyses that may be useful for others seeking to work on these metagenomes. The data availability statement has been updated in the new version of the manuscript.
Reviewer 2 Report
Comments and Suggestions for Authors
Microorganisms
Manuscript Draft
Manuscript Number: 3350654
Title: Diversity and Distribution of hydrocarbon-degrading genes in the cold seeps from the Mediterranean and Caspian Seas
Article Type: Research article
General Comments on MDPI Questions that Reviewers must answer:
- Is the manuscript clear, relevant for the field and presented in a well-structured manner?
The manuscript is written clearly, is well-structured, and is relevant to the field since it analyzes both aerobic and anaerobic microbial communities in marine cold seeps in both the Mediterranean Sea and the Caspian Sea. These microbial communities have the potential to break down hydrocarbons. This has the potential to help remediate oil spills (e.g., Deep Water Horizon) if these microorganisms can be leveraged to break down the hydrocarbons in such oil spills to mitigate adverse environmental impacts. Given the potential importance of such application, this research warrants publication. However, there needs to be another sub-section added 4.2. to the Discussion section specifically on how the research results APPLY to the development and/or use of the microbes studied to help remediate oil spills.
In addition to this major substantive edit, the following minor edits and clarifications need to be made as well:
1) Aside from the first keyword, do not capitalize words. Also, please list in alphabetical order.
2) Please use Word Find and Replace function to add a blank space in front of every [#] in text citation.
3) Please add a paragraph at the end of the Introduction section that goes over the goal(s) and objective(s) of the research.
4) Paragraphs are by definition at least 3 sentences with a topic sentence followed by at least 2 supporting sentences. Please make sure this is the case throughout the manuscript.
5) Figure 1 needs to be referred to in the writing which is presumably somewhere in L175-185. Please move Figure 1 after this reference.
6) Sub-header titles need to be in the following format which capitalizes all major words (for the most part, this has been done): 2.1. Sample Collection and DNA Extraction. Please change to this numerical format throughout the rest of the manuscript.
7) The sub-headers on L156, L198, and L205 needs to be more clear. What does 16. S mean exactly?
8) On L305, Figure 1 should not be in bold.
9) Figure 1 could be improved if Cas_Sea was written out as Caspian Sea and Med_Sea was written out as Mediterranean Sea. Figures and tables should be “stand alone” in understanding without the reader having to refer back to the writing to understand what is being shown
10) On L359, Figure 3 should not be in bold.
11) For Figures 3, 5, 6 and 7, either increase the font size of the words/numbers or put the figures (a) and (b) on top of each other.
12) Please add at a short paragraph at the end of the Conclusions on how future research can expand upon the current work.
13) Add a blank row after L704.
14) Add a blank row after L724.
15) In the References, all journal article page ranges need to be followed by a period and not a comma.
- Are the cited references mostly recent publications (within the last 5 years) and relevant? Does it include an excessive number of self-citations?
A little over one-fourth of the cited references have been published within the last 5 years and appear relevant to the research topic. There are no excessive self-citations.
- Is the manuscript scientifically sound and is the experimental design appropriate to test the hypothesis?
The manuscript is scientifically sound and the experimental design is appropriate. The research goal and objectives need to be more clearly stated in the last paragraph of 1. Introduction.
- Are the manuscript’s results reproducible based on the details given in the methods section?
The manuscript’s experimental results are reproducible based on what is described in 2. Materials and Methods.
- Are the figures/tables/images/schemes appropriate? Do they properly show the data? Are they easy to interpret and understand? Is the data interpreted appropriately and consistently throughout the manuscript? Please include details regarding the statistical analysis or data acquired from specific databases.
Please see my previous edits for figures. It is surprising that there is no experiment data presented in table(s).
- Are the conclusions consistent with the evidence and arguments presented?
The Conclusions are consistent with the evidence and arguments presented. Please add a short paragraph on how future research can improve upon the current work at the end of the Conclusions section.
- Please evaluate the data availability statements to ensure it is adequate.
The Data Availability Statement and the Ethics Statement are both fine.
Author Response
Summary
Thank you very much for taking the time to review this manuscript. Please find the detailed responses below and the corresponding revisions/corrections highlighted/in track changes in the re-submitted files.
Comment 1:
The manuscript is written clearly, is well-structured, and is relevant to the field since it analyzes both aerobic and anaerobic microbial communities in marine cold seeps in both the Mediterranean Sea and the Caspian Sea. These microbial communities have the potential to break down hydrocarbons. This has the potential to help remediate oil spills (e.g., Deep Water Horizon) if these microorganisms can be leveraged to break down the hydrocarbons in such oil spills to mitigate adverse environmental impacts. Given the potential importance of such application, this research warrants publication. However, there needs to be another sub-section added 4.2. to the Discussion section specifically on how the research results APPLY to the development and/or use of the microbes studied to help remediate oil spills.
Response 1 : Thank you for your helpful suggestion. We have added a new subsection, “Applications in Bioremediation” line 680 – 691 to the Discussion. This section explains how the microbes and hydrocarbon-degrading pathways identified in our study can help clean up oil spills in different environments. We appreciate your input, which has improved the practical relevance of our work.
Comment 2 : Aside from the first keyword, do not capitalize words. Also, please list in alphabetical order.
Response 2 : Thank you for the suggestion. We have corrected the keywords to list them alphabetically and ensured that only the first word is capitalized, except for proper nouns.
Comments 3 : Please use Word Find and Replace function to add a blank space in front of every [#] in text citation.
Response 3 : Thank you for pointing this out. We have used the Find and Replace function to ensure that a blank space has been added before every in-text citation throughout the manuscript.
Comment 4 : Please add a paragraph at the end of the Introduction section that goes over the goal(s) and objective(s) of the research.
Response 4 : Thank you for your suggestion. We have added a paragraph at the end of the Introduction section that outlines the goals and objectives of our research, highlighting the focus on microbial diversity, hydrocarbon-degrading genes, and their potential applications in bioremediation. This addition provides clarity on the purpose and scope of the study.
Comments 5 : Paragraphs are by definition at least 3 sentences with a topic sentence followed by at least 2 supporting sentences. Please make sure this is the case throughout the manuscript.
Response : Thank you for your feedback. I have carefully reviewed the manuscript and ensured the overall structure and flow are consistent. I have made necessary changes to some paragraphs to improve clarity and remove short paragraphs.
Comments 6: Figure 1 needs to be referred to in the writing which is presumably somewhere in L175-185. Please move Figure 1 after this reference.
Response 6 : Thank you for your feedback. I have added references to Figure 1 in the relevant section of the manuscript to ensure it is properly contextualized L178 -L180.
Comments 7 : Sub-header titles need to be in the following format which capitalizes all major words (for the most part, this has been done): 2.1. Sample Collection and DNA Extraction. Please change to this numerical format throughout the rest of the manuscript.
Response 7 : Thank you for your feedback. I have reviewed the manuscript and updated all sub-header titles to follow the requested numerical format, ensuring consistency throughout the document.
Comments 8 : The sub-headers on L156, L198, and L205 needs to be more clear. What does 16. S mean exactly?
Response 8 : Thank you for your comment. We reviewed the manuscript and found "16 S" written on lines 186, 198, and 205. We have corrected it to "16S" to ensure accuracy and consistency.
Comments 9 : On L305, Figure 1 should not be in bold.
Response 9 : Thank you for your comment. I have removed the bold formatting from "Figure 1" on Line 305.
Comments 10 : Figure 1 could be improved if Cas_Sea was written out as Caspian Sea and Med_Sea was written out as Mediterranean Sea. Figures and tables should be “stand alone” in understanding without the reader having to refer back to the writing to understand what is being shown
Response 10 : Thank you for the suggestion. I have updated Figure 1 to replace "Cas_Sea" with "Caspian Sea" and "Med_Sea" with "Mediterranean Sea." The figure has been revised to ensure it is standalone and clear.
Comments 11 : On L359, Figure 3 should not be in bold.
Response 11 : Thank you for your comment. I have removed the bold formatting from "Figure 3" on Line 359.
Comments 12 : For Figures 3, 5, 6 and 7, either increase the font size of the words/numbers or put the figures (a) and (b) on top of each other.
Response 12 : Thank you for the feedback. I have updated Figures 3, 5, 6, and 7 by increasing the font size of the words and numbers to enhance readability. Additionally, the layout of the figures has been adjusted where necessary to ensure clarity.
Comments 13 : Please add at a short paragraph at the end of the Conclusions on how future research can expand upon the current work.
Response 13 : Thank you for the suggestion. I have added a short paragraph at the end of the Conclusions section outlining how future research can build upon the current work.
Comments 14 : Add a blank row after L704.
Response 14 : Thank you for your comment. I have added a blank row after Line 704, as requested.
Comments 15 : Add a blank row after L724.
Response 15 : Thank you for your comment. I have added a blank row after Line 724, as requested.
Comments 16 : In the References, all journal article page ranges need to be followed by a period and not a comma.
Response 16 : Thank you for pointing this out. I have manually corrected the formatting in the References section to ensure that all journal article page ranges are followed by a period. This adjustment was made because Zotero did not automatically add the period, and the reason for this issue is unclear.